# Measuring Impacts of Urban Environmental Elements on Housing Prices Based on Multisource Data—A Case Study of Shanghai, China

**Liujia Chen** [1,2,3], **Xiaojing Yao** [1,3,*], **Yalan Liu** [1,3], **Yujiao Zhu** [4], **Wei Chen** [4], **Xizhi Zhao** [5] **and Tianhe Chi** [1,3]

1   Airspace Information Research Institute, Chinese Academy of Sciences, Beijing 100094, China;
    chenliujia2013@gmail.com (L.C.); liuyl@radi.ac.cn (Y.L.); chith@126.com (T.C.)
2   University of Chinese Academy of Science, Beijing 100049, China
3   Lab of Spatial Information Integration, Institute of Remote Sensing and Digital Earth, Chinese Academy of
    Sciences, Beijing 100101, China
4   School of Geosciences & Surveying Engineering, China University of Mining & Technology, Beijing 100083,
    China; 15621341509@163.com (Y.Z.); chenw@cumtb.edu.cn (W.C.)
5   Research Center of Government Geographic Information System, Chinese Academy of Surveying and
    Mapping, Beijing 100830, China
*   Correspondence: yaoxj@aircas.ac.cn; Tel.: +86-1860-043-0682

**Abstract:** Diverse urban environmental elements provide health and amenity value for residents. People are willing to pay a premium for a better environment. Thus, it is essential to assess the benefits and values of these environmental elements. However, limited by the interpretability of the machine learning model, existing studies cannot fully excavate the complex nonlinear relationships between housing prices and environmental elements, as well as the spatial variations of impacts of urban environmental elements on housing prices. This study explored the impacts of urban environmental elements on residential housing prices based on multisource data in Shanghai. A SHapley Additive exPlanations (SHAP) method was introduced to explain the impacts of urban environmental elements on housing prices. By combining the ensemble learning model and SHAP, the contributions of environmental characteristics derived from street view data and remote sensing data were computed and mapped. The experimental results show that all the urban environmental characteristics account for 16 percent of housing prices in Shanghai. The relationships between housing prices and two green characteristics (green view index from street view data and urban green coverage rate from remote sensing) are both nonlinear. Shanghai's homebuyers are willing to pay a premium for green only when the green view index or urban green coverage rate are of higher value. However, there are significant differences between the impacts of the green view index and urban green coverage rate on housing prices. The sky view index has a negative influence on housing prices, which is probably because the high-density and high-rise residential area often has better living facilities. Residents in Shanghai are willing to pay a premium for high urban water coverage. The case of Shanghai shows that the proposed framework is practical and efficient. This framework is believed to provide a tool to inform the decisions of housing buyers, property developers and policies concerning land-selling and buying, property development and urban environment improvement.

**Keywords:** street view; remote sensing; urban environmental elements; ensemble learning; green view; sky view; building view; SHAP

---

## 1. Introduction

Urban green spaces, sky and other urban environmental elements can significantly affect the quality of urban life [1,2]. Various studies have shown that urban environmental elements have a significant influence on people's physical and mental health. For instance, urban green spaces have multiple ecological benefits, including air purification [3,4], climate regulation [5], carbon storage [6] and noise reduction [7]. In addition, green spaces provide plenty of spaces for pressure releasing and, consequently, positively affect mental health [8–10]. Higher levels of sky view visibility were associated with lower psychological distress [11]. Contrary to green and sky, high-rise buildings make people feel stressed [12]. With rapid urbanization and improvement of living standards, increasing concern about the quality and quantity of urban environmental elements has grown all over the world. Many people display a marked preference for natural over built environmental elements [13]. This preference is often shown by the housing choices of consumers in the residential housing market. People are willing to pay extra for a home with more natural environmental elements [14].

The explanatory variables of housing prices have been widely discussed in the housing literature. Bangura, Lee and Al-Masum discussed the ability of market fundamentals in explaining housing prices from the macroeconomic perspective [15–17], while Trojanek and Yamagata examined the importance of housing attributes in explaining housing prices from the microeconomic perspective [18,19]. In recent years, a great deal of research has studied the impacts of environmental elements on housing price. For instance, a house with a water view could attract a premium of 8%–10% in the Netherlands [20]. In Guangzhou, the view of green spaces and proximity to water bodies can lead to a considerable increase in house price, contributing at 7.1% and 13.2%, respectively [1]. An additional street tree increases a house's monthly rental price by $21.00 in Portland, Oregon, USA [21]. In Singapore, vegetation had positive effects on housing prices, accounting for 3% of a property's value [22]. On the contrary, both street and building views would depress housing price, with the influence of street view more significant than building view in Hong Kong [23]. However, most of the existing studies analyze the impacts of urban environmental elements on housing prices by using field survey data [1,24] and satellite remote sensing data [25,26]. Field survey data is time-consuming and hard to be applied in large-scale studies. Satellite remote sensing data is limited by an overhead view perspective and spatiotemporal resolution. Street view images bring a new opportunity to obtain urban environmental elements. This type of data has the advantages of easy obtaining, wide coverage and high spatial resolution. More importantly, street view images represent a horizontal view perspective, which is closer to the general population's perception of urban environmental elements. The rapid development of computer vision provides an efficient method for the information extraction of street view images. In this context, a great number of studies have been conducted to measure street-level green [27], estimate the spatiotemporal patterns of urban mobility [28], examine the relationship between street view and perceived safety [29] and assess the visual quality of urban environment [30] Therefore, in this study, street view data is used to evaluate the relationship between urban environmental elements and housing prices.

Most of the existing studies conducted on the impacts of urban environmental elements on housing prices used the hedonic pricing model (HPM) as the research method. This method assumes that real estate is heterogeneous and three types of characteristics have significant impacts on housing price, namely structure, neighborhood and location characteristics [31,32]. In empirical research, HPM mainly has three forms, including linear models [24,30], semi-log models [1] and double-log models [33]. However, most studies combine linear regression with HPM to interpret the impact of different independent variables [34,35]. No matter which form HPM is, only the log transformation of independent variables or dependent variables is performed for reducing the heteroscedasticity of the model. Therefore, the hedonic model is limited to revealing the complicated nonlinear relationships between housing prices and a variety of potential determinants [36]. In addition, the combination of linear regression and HPM explains the impact of a housing characteristic on housing prices by the value of this characteristic and the same corresponding regression coefficients of the regression equation.

Thus, this method could not reveal the spatial variations of the contribution of each characteristic. To address these problems, we propose an analytical framework which combines ensemble learning and SHapley Additive exPlanations (SHAP). By combing the individual machine learning methods to form a new classifier, ensemble learning algorithms such as Random Forest Regression (RFR) and XGBoost Regression (XGBoost) achieve better performance than any of the individual ones [37]. Compared to traditional methods, these ensemble learning algorithms show obvious advantages in three aspects: (1) capability to capture nonlinear relationships, (2) high prediction accuracy and (3) capability to capture high-order interactions between inputs. Recent urban housing prices studies have shown the advantage of ensemble learning algorithms over traditional methods [28,38]. Hu compared the performance of six machine learning algorithms in monitoring housing rental prices and found that ExtraTrees and RFR get better results [39]. However, because the nature of ensemble learning models are not interpretable models, almost all of these studies only range the importance when measuring the impacts of a housing characteristic on housing prices. It is hard to analyze the contribution of each characteristic to the housing price. SHAP, which is based on the game theoretically optimal Shapley values, falls into this specific scope and provides a new opportunity for solving this problem. Unlike methods that provide a specific global predictor, the SHAP framework provides an explanation of the model overall behavior in the form of particular feature contributions. Thus, this method can be used to explain the spatial variations of the contribution of each characteristic and the complex nonlinear relationships between each characteristic and housing prices. SHAP is becoming an increasingly popular tool to interpret natural and social phenomena [40,41].

In brief, the main contributions of this study are as follows. (1) Considering the perception of the urban environment from the horizontal view perspective, which could be easier for ordinary people to understand, street view data is used to calculate the environmental characteristics. (2) Tree-based ensemble learning regression algorithms are employed to model the housing prices and a method for explanting these ensemble learning models—SHAP is introduced to interpret the relationships between urban environmental elements and housing prices. By combining tree-based ensemble learning regression algorithms and the SHAP model, the complex and nonlinear relationships between most of the environmental elements and housing prices are revealed, which is more elaborate than the results of previous studies. (3) SHAP models are employed for the geospatial analysis of housing prices. The spatial distribution of SHAP for five environmental characteristics were mapped to improve the understanding of the spatial variations of each urban environmental characteristic's contribution. (4) The impacts of the green view index from street view data and green coverage rate from remote sensing data are compared in this study. The difference impacts of the same urban environmental elements from different observation perspectives provide new insights into urban environment research.

The remainder of this paper is organized as follows. Section 2 introduces the study area, data and methods used in this study. Section 3 presents the research results and discusses the reasons behind these results and suggestions for future work. Section 4 provides a conclusion of our study.

## 2. Data and Methods

### 2.1. Study Area

Shanghai, one of the financial, trade, economic and shipping hubs in the world, is located on China's east coast. Since the implementation of housing reforms that transformed the housing system from an administrative allocation model to a market mechanisms model in 1980, housing prices in Shanghai have ballooned over the years [42]. At present, Shanghai has become one of the most expensive housing markets, with a large number of housing transactions. The area within the outer ring road, which has a population density of 17,070 per square kilometer, is regarded as the central city of Shanghai [43]. With such a high-density population, a large number of housing transactions occur

in this area. Therefore, an empirical analysis in the area within the outer ring road can supply essential references for relevant studies. The study areas in this paper are shown in Figure 1a.

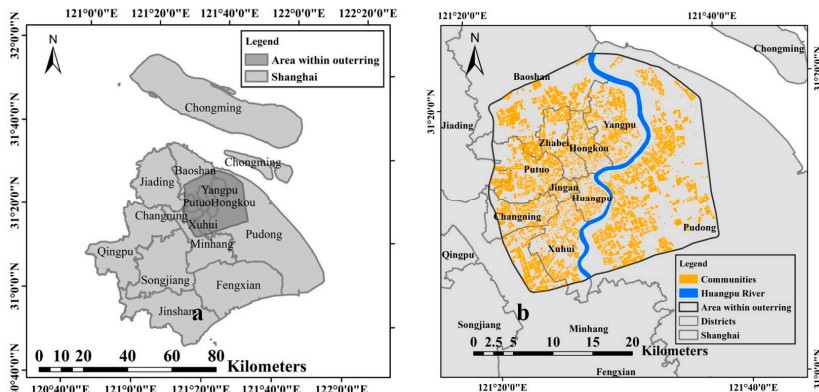

**Figure 1.** Location map of the study area: the area within the outer ring road of Shanghai (**a**) and the distribution of communities (**b**).

*2.2. Overall Methodological Framework*

Figure 2 presents the overall methodological framework, which follows three major steps to complete the analysis. First, multisource data were gathered and cleaned to extract the housing prices and corresponding characteristics at the community level. Second, we used these housing prices and characteristics to select the most appropriate machine learning model. Finally, by inputting the selected machine learning model and the characteristics of the communities into the SHapley Additive exPlanations model (SHAP), the SHAP value of these characteristics were computed to analyze the global importance of the characteristics and the contribution of urban environmental characteristics.

*2.3. Characteristics Extraction*

In China, taking the form of a gated residential area, a community is regarded as a basic management unit of urban planning [44]. In addition, houses located in the same community share a similar urban environment. Therefore, we chose communities as the basic analytical units in this paper. By crawling Baidu Maps, we obtained 7043 community boundaries in the study area (Figure 1b). All the housing characteristics involved in this study were transformed to the same community units for further study.

2.3.1. Housing Price

In this study, based on a web crawler, we collected the historical transactional data of preowned houses from Lianjia.com in 2018. There were four steps in the processing of preowned houses transaction data. First, a web crawler was used to download the historical transaction data of preowned houses, which occurred in 2018 from Lianjia.com. The transaction data recorded a number of housing attributes, including address, community name, total price, total area, price per square meter, elevator and construction time of building. Then, the collected data were cleared for (1) records whose spatial position are outside the area within the outer ring road; (2) records with missing important attributes, such as "elevator" and "construction time of building" and (3) repeated records. Finally, the price per square meter was averaged for each community. As a result of housing transactional data processing, we obtained 2547 study units with observed historical transactional data. Figure 3 presents the spatial distribution of the community-level housing prices.

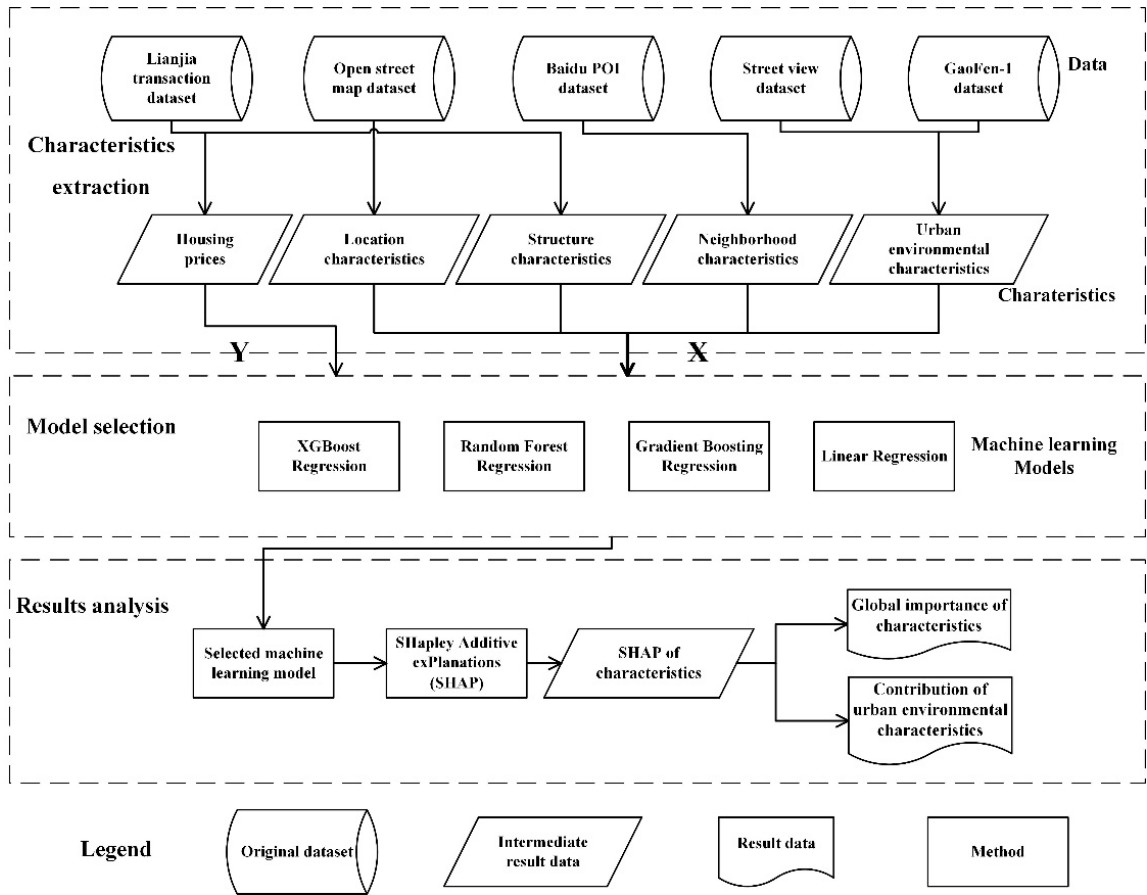

**Figure 2.** The overall methodological framework.

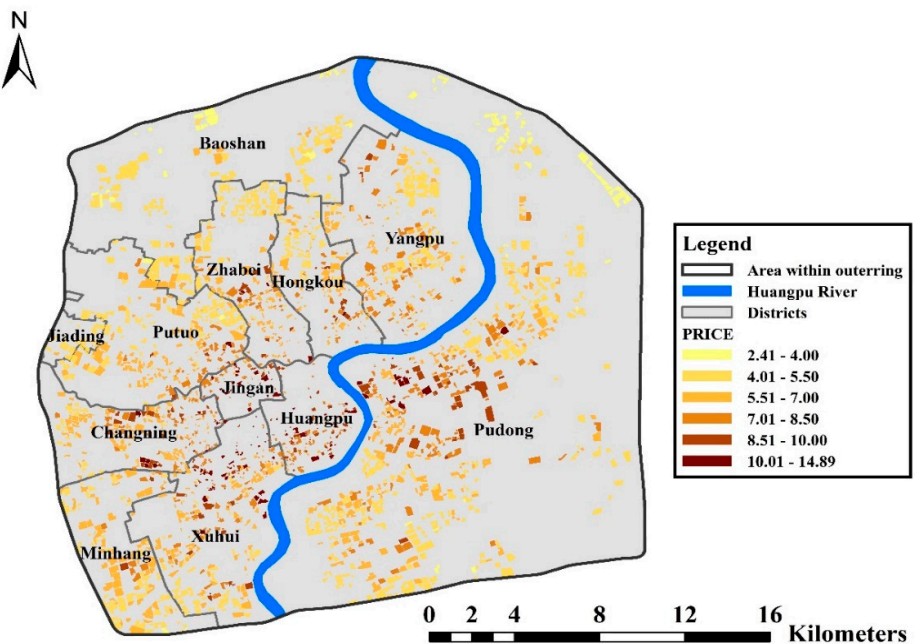

**Figure 3.** Spatial distribution of community-level housing prices.

### 2.3.2. Urban Environmental Characteristics from Street View Data

Street view data represents the urban environmental elements from a horizontal view perspective, which is closer to the general population's perception and could be easier for ordinary people to understand. Therefore, street view data was employed to measure the impacts of urban environmental elements on housing prices in this study.

The process for computing urban environmental characteristics from street view data involves three steps: street view data crawling, environmental elements extraction and characteristic calculation.

First, we selected main roads within the area of the outer ring road based on Shanghai's OpenStreetMap dataset. After that, the centerlines of these main roads were extracted, and then, we got street view sample sites along the centerlines at 50-m intervals. Each sample site was represented by a panoramic street view image. Finally, by inputting the spatial coordinate of sample sites into a Baidu static picture API, we crawled 84,520 panoramic street view images, which were acquired on August and September, 2017. Each of them has a size of 1024 by 290 pixels.

In this study, we mainly focused on three horizontal view environmental elements, including green, sky and building. Each of the elements was defined as the ratio of pixels associated with the specific element to the total pixels in a street view image. Specifically, the values of the green view index (GVI), the sky view index (SVI) and the building view index (BVI) were calculate by following equations:

$$\text{GVI} = \frac{\text{Pixels}_{\text{green}}}{\text{Pixels}_{\text{total}}} \tag{1}$$

$$\text{SVI} = \frac{\text{Pixels}_{\text{sky}}}{\text{Pixels}_{\text{total}}} \tag{2}$$

$$\text{BVI} = \frac{\text{Pixels}_{building}}{\text{Pixels}_{\text{total}}} \tag{3}$$

The rapid development of computer vision, especially the deep convolutional neural network (DCNN), provides a new method for the information extraction of images. The state-of-the-art DCNNs such as SegNet [45], PSPNet [46] and DeepLabv3 [47] were employed for image semantic segmentation and exhibited an outstanding performance in image interpretation [27]. In this study, DeepLabv3, one of the most popular image semantic segmentation models, was applied to extract street-level environmental elements at the pixel level. Figure 4 shows the flow charts of the street view images' semantic segmentation. DeepLabv3 was first pretrained using the Cityscapes dataset and was then used to segment the street view data for extracting green space, sky and building. DeepLabv3 combines an atrous convolution with upsampled filters to solve the problem of segmenting objects at multiple scales. The performance of this model outperformed the state-of-the-art models on the PASCAL VOC 2012 semantic image segmentation benchmark [47]. The Cityscapes dataset was employed to pretrain the DeepLabv3 model. Cityscapes is a large-scale dataset containing a variety of stereo video sequences at street level from 50 different cities. Five-thousand of these images have high-quality pixel-level labeling [48]. DeepLabv3 achieved 81.3% accuracy on the Cityscapes dataset. The configuration of the hardware devices used in this study were an Intel i7-8700k CPU, a NVIDIA 1080ti graphics card with 12GB video memory and 32 GB physical memory. The operation system of the computer is 64-bit Windows 10 Professional.

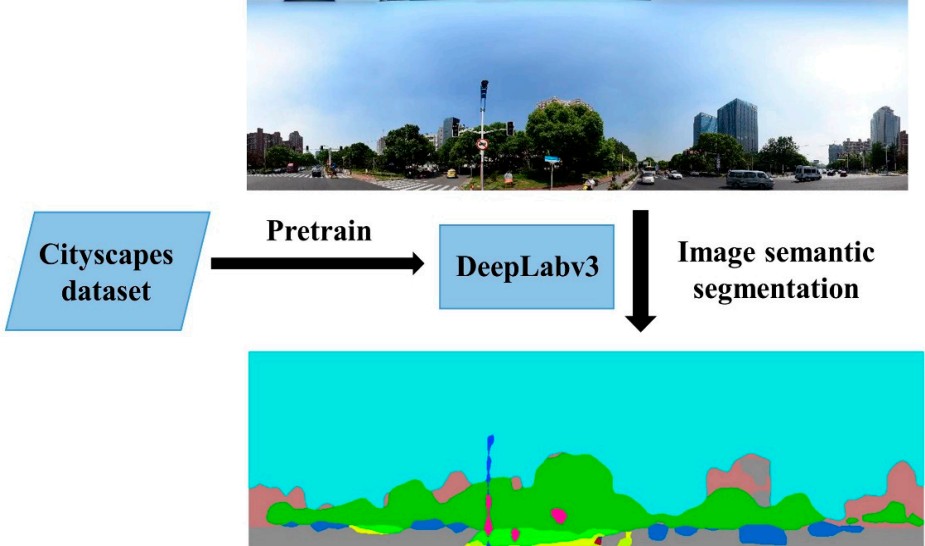

**Figure 4.** The flow chart of the street view images' semantic segmentation.

For the characteristics calculations, the GVI, SVI and BVI for each community with a 400 m radius buffer were averaged to obtain environmental characteristics at the community level. The reason why we chose 400 m is that the square root of the average area of Shanghai's communities is about 400 m, and the scope of citizens' public lives has been well-covered by this buffer. The willingness to buy a house are influenced not only by the view from their apartment but also by the view from their public life. After the calculation, there were 115 sample sites per community.

### 2.3.3. Urban Environmental Characteristics from Remote Sensing Data

To compare the urban environmental characteristics derived from street view data with remotely sensed characteristics, GaoFen-1 data were used to calculate the urban green coverage rate (UG) and urban water coverage rate (UW). Four GaoFen-1 images used in this paper were acquired on April and May, 2015, all of which consisted of four multispectral bands at an 8 m spatial resolution and one panchromatic band at a 2 m spatial resolution. The supervised classification was conducted to extract green and water by the support vector machine (SVM) tool in ENVI 5.3. Specifically, 80 green water samples and 80 water samples were randomly selected by visual interpretation. For each type of land cover, 50 samples were chosen for the training classification model and 30 samples for testing. The classification performance was assessed by a confusion matrix of test samples. The total precision was 96.75%, and the Kappa coefficient was 0.9578. The classification results are shown in Figure 5. For the characteristics calculations, the UG and UW for each community with a 400 m radius buffer were averaged to obtain the environmental characteristics from remote sensing data at the community level.

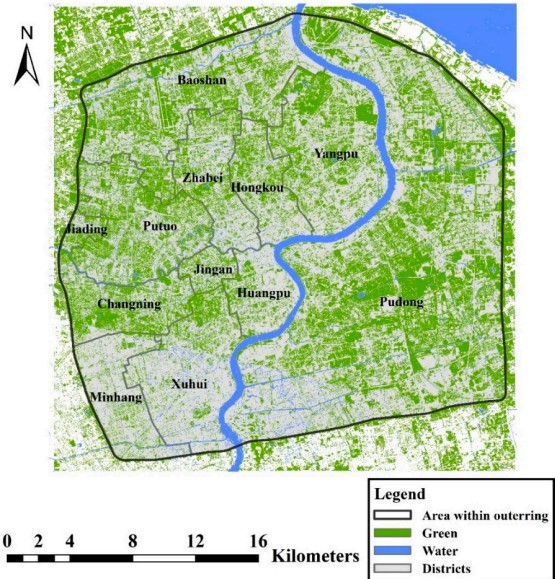

**Figure 5.** The classification results of green and water within the study area based on GaoFen-1 images.

### 2.3.4. Other Characteristics

In light of the attributes of preowned house transaction data and the spatial scale of study units, the year of construction (YEAR), average construction area of the apartment (AREA), plot ratio (PR) and whether the elevator is available (EL) were selected as structure characteristics. The variable of AREA should be introduced, because that area significantly affects the housing prices in Chinese megacities. Specifically, small houses often have a higher price per square meter because of lower total prices. Big houses (AREA > 200 m$^2$) also have a higher price per square meter due to better facilities and management. EL in original transaction data is a dummy variable. If the elevator is available in the apartment building, the value is 1; otherwise, the value is 0. For PR, the plot ratio of a community was obtained by dividing the gross floor area of the building by the area of the total community area on which the building was erected. In this study, this variable was calculated by the building footprint and Baidu community data.

For location characteristics, the distance to the city center (C_DIS), the city employment center (EC_DIS), river (R_DIS) and the Huangpu River (HPR_DIS) were chosen. In detail, the Bund was selected as the city center of Shanghai, and the employment center identified by Sun was used in this study [49]. The reason why the HPR_DIS was chosen is that the distance from each neighborhood centroid to the Huangpu River notably affects residential housing prices. The housing prices decrease with the increase of the distance [30].

For neighborhood characteristics, the variables which measured the accessibility to bus stations, subway stations, primary schools and first-class hospitals at grade 3 (hospitals with high-quality facilities and services) were included in our study. Using the points of interest (POI) data collected from the Baidu Map, the distance from each community to its nearest facility and the number of facilities within a specified distance were calculated. Specifically, 500 m and 1000 m were selected as the distance threshold in the density calculation, considering the 15-min community life circle proposed by the Chinese government.

General descriptive statistics of the selected housing characteristics are shown in Table 1.

**Table 1.** General descriptive statistics of the housing characteristics.

| Variables | Description | Mean | Standard Deviation | Range |
|---|---|---|---|---|
| **Dependent variable** | | | | |
| PRICE | Transaction price (10,000 RMB/m$^2$) | 6.347 | 1.678 | 2.413–14.894 |
| **Location characteristics** | | | | |
| C_DIS | Distance to the city center (10 km) | 0.792 | 0.331 | 0.046–1.650 |
| EC_DIS | Distance to the city employment centers (10 km) | 0.295 | 0.188 | 0–1.092 |
| R_DIS | Distance to the river (10 km) | 0.278 | 0.198 | 0.02–1.138 |
| HPR_DIS | Distance to the Huangpu River (10 km) | 0.420 | 0.283 | 0.003–1.311 |
| **Structure characteristics** | | | | |
| YEAR | 2019 minus the construction time of building | 21.622 | 9.121 | 2–106 |
| AREA | Average construction area in the apartment (m$^2$) | 78.788 | 38.743 | 22–346 |
| PR | Plot ratio | 2.600 | 1.234 | 0–13.703 |
| EL | Dummy variable, 1 if elevator is available. | 0.398 | 0.470 | 0–1 |
| **Neighborhood characteristics** | | | | |
| BUS_NEAR | Distance to the nearest bus station (km) | 0.083 | 0.091 | 0–0.996 |
| BUS_500M | Number of bus stations within 500 m | 9.894 | 3.862 | 0–25 |
| BUS_1000M | Number of bus stations within 1000 m | 30.740 | 8.887 | 4–73 |
| SUB_NEAR | Distance to the nearest subway station (km) | 0.704 | 0.548 | 0–4.167 |
| SUB_500M | Number of subway stations within 500 m | 0.577 | 0.641 | 0–3 |
| SUB_1000M | Number of subway stations within 1000 m | 1.940 | 1.297 | 0–7 |
| PRI_NEAR | Distance to the nearest primary school (km) | 0.365 | 0.298 | 0–2.259 |
| PRI_500M | Number of primary schools within 500 m | 1.611 | 1.280 | 0–7 |
| PRI_1000M | Number of primary schools within 1000 m | 4.847 | 2.769 | 0–18 |
| FH3_NEAR | Distance to the nearest first-class hospital at grade 3 (km) | 2.221 | 1.641 | 0.026–7.614 |
| FH3_500M | Number of first-class hospitals at grade 3 within 500 m | 0.154 | 0.435 | 0–3 |
| FH3_1000M | Number of first-class hospitals at grade 3 within 1000 m | 0.547 | 0.976 | 0–6 |
| **Urban Environmental characteristics** | | | | |
| GVI | Mean green view index within 400 m distance | 0.315 | 0.123 | 0–0.828 |
| SVI | Mean sky view index within 400 m distance | 0.470 | 0.124 | 0–0.798 |
| BVI | Mean building view index within 400 m distance | 0.117 | 0.071 | 0–0.403 |
| UG | Urban green coverage rate | 0.381 | 0.154 | 0.020–0.755 |
| UW | Urban water coverage rate | 0.025 | 0.032 | 0–0.380 |

### 2.4. Ensemble Learning Algorithms

The relationships between housing prices and housing characteristics is complex and nonlinear. By combing a bunch of individual models and averaging the individual result, ensemble learning algorithms are more flexible and less data-sensitive. Thus, ensemble learning algorithms are suitable for modeling housing prices. The most commonly used ensemble learning methods are bagging and boosting. The difference between these two methods is that bagging methods train a number of individual models by a random subset of train data in a parallel way while boosting methods train models in a sequential way for learning mistakes made by the previous model. In this study, three

tree-based ensemble learning algorithms and linear regressions were employed to model housing prices for selecting the algorithm. Random forest regression (RFR) uses bagging as the ensemble method and decision tree as the individual model. Since RFR trains each tree independently and uses random subsets from the training set, this method is less likely to overfit [50]. Gradient boosting regression (GBR), a boosting model, builds trees one at a time, where each new tree aims to correct errors in the predictions made by all previous trees [51]. Achieving high accuracy in a wide range of practical applications, XGBoost is an optimized distributed gradient boosting method based on ensembles of classification and regression trees (CARTs) [52]. This method provides a parallel tree-boosting to solve problems in a fast and accurate way.

Different algorithms have their own strengths and weaknesses. Therefore, to choose the optimal ensemble learning algorithms, we compared their performances in the explanation of housing prices. In detail, the regression performances of the four algorithms were measured by five common metrics, including explained variance score, mean absolute error (MAE), mean squared error (MSE), median absolute error (MedAE) and the coefficient of determination ($R^2$):

$$\text{explained variance}(y, \hat{y}) = 1 - \frac{\text{Var}\{y - \hat{y}\}}{\text{Var}\{y\}} \tag{4}$$

$$\text{MAE}(y, \hat{y}) = \frac{1}{n} \sum_{i=0}^{n-1} |y_i - \hat{y}_i| \tag{5}$$

$$\text{MSE}(y, \hat{y}) = \frac{1}{n} \sum_{i=0}^{n-1} (y_i - \hat{y}_i)^2 \tag{6}$$

$$\text{MedAE}(y, \hat{y}) = \text{median}(|y_1 - \hat{y_1}|, \dots, |y_n - \hat{y_n}|) \tag{7}$$

$$R^2(y, \hat{y}) = 1 - \frac{\sum_{i=1}^{n} (y_i - \hat{y})^2}{\sum_{i=1}^{n} (y_i - \overline{y})^2} \tag{8}$$

where y and $\hat{y}$ are the true housing price and the estimated housing price, Var is Variance, n denotes the total number of communities, $y_i$ and $\hat{y}_i$ represent the predicted housing price of the i-th community and the corresponding true value, $\hat{y}_n$ means the predicted housing price of the n-th community and $\overline{y}$ is the mean true housing price.

All the experiments in this study were performed by using a scikit-learn and XGBoost Python package. For the hyperparameter tuning and the accuracy evaluation, we chose a 10-fold cross-validation, which is a common method for performance validation.

*2.5. Shapley Additive Explanations*

Proposed by Lundberg and Lee, SHapley Additive exPlanations (SHAP) is a method to explain the prediction of a specific instance by calculating the contribution of each feature to the prediction [53]. The SHAP method computes Shapley values from coalitional game theory. The Shapley value of a feature value is its contribution to the output value, weighted and summed over all possible feature value combinations. The value of the j-th feature contributed $\phi_j$ was calculated as follow:

$$\phi_j(\text{val}) = \sum_{S \subseteq \{x_1, \dots, x_p\} \smallsetminus \{x_j\}} \frac{|S|!(p - |S| - 1)!}{p!} (\text{val}(S \cup \{x_j\}) - \text{val}(S)) \tag{9}$$

where p is the number of features, S represents a subset of the features used in the model, x denotes the vector of feature values of an instance to be explained and val(S) means the prediction for feature values in set S.

The advantages of SHAP include: (1) global interpretability—the collective SHAP value is able to identify the positive or negative relationship for each variable with the target and (2) local interpretability—each feature of an instance gets its own corresponding SHAP values. Traditional variable importance algorithms are limited to obtain the results across the entire population but not on each individual instance. Meanwhile, we can also measure the global importance of characteristics by computing the absolute Shapley values per characteristic:

$$I_j = \sum_{i=1}^{n} |\phi_j^{(i)}| \tag{10}$$

where $\phi_j^{(i)}$ represents the SHAP value of the j-th feature for instance i.

In this paper, we employed SHAP feature attributions, SHAP explanation force plots, SHAP summary plots and SHAP partial dependence plots and interaction plots to explore the relationships between housing prices and urban environmental elements. The XGBoost and shap Python packages were used for implementing SHAP.

## 3. Results and Discussion

### 3.1. Spatial Dstribution of Urban Environmental Characteristics

To enhance the understanding of the environmental elements of study area, we plotted the spatial distribution of five urban environmental characteristics in Figure 6. Each characteristic was mapped using seven value intervals by the natural breaks method. The average value of the green view index (GVI), sky view index (SVI), building view index (BVI), urban green coverage rate (UG) and urban water coverage rate (UW) at the community level were 0.315, 0.473, 0.117, 0.381 and 0.025, respectively. Figure 6a shows that the communities with high GVI were mainly located in the Yangpu District, Hongku District, Changning District, the northeast of Putuo District and the south of Baoshan District. Figure 6b indicates the value of SVI were the lowest in the central area and increase to the outskirts gradually, while the BVI values show the opposite pattern in Figure 6c. Figure 6d demonstrates that the UG values also increased from the central area to the outskirts gradually. From Figure 6e, we can find that the communities with high UW were mainly concentrated along the Huangpu River and the Suzhou Creek.

### 3.2. Model Selection

The multicollinearity between variables, which were measured by the variance inflation factor (VIF), and the results of the hedonic model, which was built by the linear regression model, are shown in Table 2. The VIFs of all the characteristics were lower than four, which indicated that these characteristics did not have serious multicollinearity. The performances of the ensemble learning regression algorithms and linear regression algorithms are compared in Table 3. Table 3 shows that the explained variance score ranged from 0.5023 to 0.6820, the MAE ranged from 0.6554 to 0.8509, the MSE ranged from 0.8556 to 1.3784, the MedAE ranged from 0.4848 to 0.6549 and the $R^2$ ranged from 0.4847 to 0.7045. The performances of the three ensemble methods were much better than linear regression. Among the three ensemble methods, XGBoost regression presented the best performance and was selected to be trained for interpreting the impact of urban environmental elements on housing prices.

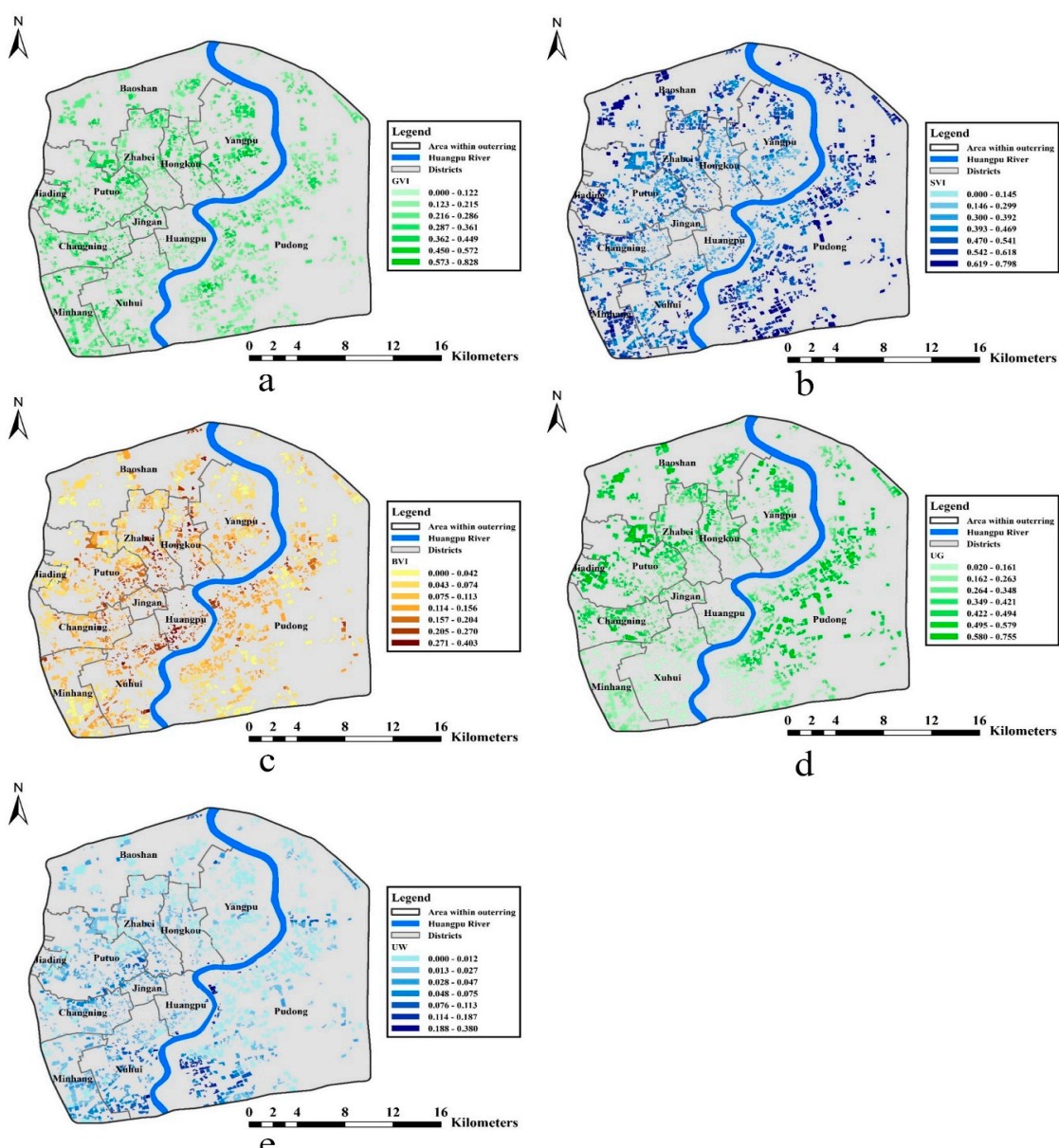

**Figure 6.** The spatial distributions of urban environmental characteristics: (**a**) green view index (GVI), (**b**) sky view index (SVI), (**c**) building view index (BVI), (**d**) urban green coverage (UG) and (**e**) urban water coverage (UW).

**Table 2.** The unstandardized coefficients, standard error and variance inflation factor (VIF) values of variables.

| Variables | Unstandardized Coefficients | Standard Error | VIF |
|---|---|---|---|
| Constant | 8.342 | 0.368 | |
| **Location characteristics** | | | |
| C_DIS | −1.196 *** | 0.123 | 3.191 |
| EC_DIS | −1.771 *** | 0.168 | 1.932 |
| R_DIS | 0.021 | 0.154 | 1.781 |
| HPR_DIS | −0.236 ** | 0.102 | 1.602 |
| **Structure characteristics** | | | |
| YEAR | −0.042 *** | 0.004 | 2.343 |
| AREA | 0.003 *** | 0.001 | 1.987 |
| PR | −0.161 *** | 0.022 | 1.453 |
| EL | 0.467 *** | 0.073 | 2.277 |
| **Neighborhood characteristics** | | | |
| BUS_NEAR | −0.075 | 0.282 | 1.255 |
| BUS_500M | $-9.972 \times 10^{-5}$ | 0.009 | 2.577 |
| BUS_1000M | 0.002 | 0.004 | 2.788 |
| SUB_NEAR | −0.094 ** | 0.060 | 2.115 |
| SUB_500M | 0.122 | 0.048 | 1.856 |
| SUB_1000M | 0.156 *** | 0.025 | 2.100 |
| PRI_NEAR | 0.356 *** | 0.098 | 1.634 |
| PRI_500M | 0.069 *** | 0.026 | 2.061 |
| PRI_1000M | 0.025 * | 0.013 | 2.497 |
| FH3_NEAR | −0.077 *** | 0.023 | 2.824 |
| FH3_500M | −0.005 | 0.067 | 1.658 |
| FH3_1000M | 0.180 *** | 0.034 | 2.153 |
| **Urban Environmental characteristics** | | | |
| GVI | 0.710 ** | 0.329 | 3.143 |
| SVI | −1.235 *** | 0.317 | 2.964 |
| BVI | 0.088 | 0.539 | 2.838 |
| UG | −0.053 | 0.191 | 1.652 |
| UW | 6.494 *** | 0.856 | 1.475 |

* Indicates significance at the 10% level, ** indicates significance at the 5% level and *** indicates significance at the 1% level.

**Table 3.** Performance of linear regression algorithms and three ensemble learning regression algorithms. MAE: mean absolute error, MSE: mean squared error, MedAE: median absolute error and $R^2$: coefficient of determination.

| | Linear Regression | XGBoost Regression | Random Forest Regression | Gradient Boosting Regression |
|---|---|---|---|---|
| Explained variance score | 0.5023 | **0.6820** | 0.6398 | 0.5887 |
| MAE | 0.8509 | **0.6554** | 0.6918 | 0.7697 |
| MSE | 1.3784 | **0.8556** | 0.9703 | 1.1340 |
| MedAE | 0.6549 | **0.4848** | 0.4891 | 0.5876 |
| $R^2$ | 0.4847 | **0.7045** | 0.6306 | 0.5747 |

In order to investigate whether urban environmental characteristics from the horizontal view and from the overhead view will affect the housing prices, we estimated the $R^2$ of four additional models: model 1 only with location, structure and neighborhood characteristics; three horizontal view

urban environmental characteristics (GVI, SVI and BVI) were added to model 2 based on model 1; two overhead view urban environmental characteristics (UG and UW) were added to model 3 based on model 1 and model 4 included all the characteristics. As shown in Table 4, adding either horizontal view urban environmental characteristics or overhead view ones led to a significant improvement of $R^2$. Specifically, horizontal view urban environmental characteristics increased $R^2$ by 0.0249 and overhead view ones increased $R^2$ by 0.0265. Adding all the urban environmental characteristics resulted in the highest $R^2$ of 0.7045. These results suggested that both urban environmental characteristics from the horizontal view and from the overhead view can affect housing prices. The following section further analyzed the impacts of urban environmental elements on housing prices based on model 4.

**Table 4.** Model performance with difference characteristics.

|  | Model 1 | Model 2 (Model 1 + GVI + SVI + BVI) | Model 3 (Model 1 + UG + UW) | Model 4 (Model 1 + GVI + SVI + BVI + UG + UW) |
|---|---|---|---|---|
| $R^2$ | 0.6722 | 0.6971 | 0.6987 | 0.7045 |

### 3.3. Global Importance of Characteristics

In this section, we compared the global importance of all characteristics by calculating the SHAP feature importance. We run SHAP for communities based on the trained XGBoost models and got a matrix of Shapley values.

To facilitate the understanding, we took Aijian mansion as an example. Figure 7 shows characteristics each contributing to push the model output from the base value (the baseline for Shapley values is the average of all outputs) to the model output. Characteristics pushing the prices higher were shown in red; those pushing the prices lower were in blue. The baseline—the average predicted housing prices, was 6.373. The predicted price of Aijian mansion was 5.90. EC_DIS increased the price by 0.04922, while HPR_DIS decreased the price by 0.6428.

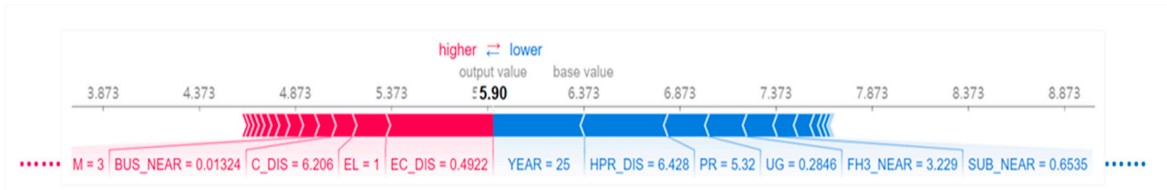

**Figure 7.** SHapley Additive exPlanations (SHAP) explanation force plots for Aijian mansion.

Based on the matrix of Shapley values, the absolute Shapley values per characteristic across the data were computed for measuring the global importance of characteristics by Formula (10). We sorted the characteristics by decreasing importance and plotted them in Figure 8. The top characteristics contributed more to the model than the bottom ones, and thus, had a greater impact on the housing prices. Overall, the four categories of the characteristics' SHAP importance could be ranked as follows: location characteristics (0.8491) > neighborhood characteristics (0.7055) > structure characteristics (0.6939) > urban environmental characteristics (0.4266). This result indicated that the location characteristics were the dominant determinants of housing prices in Shanghai. The importance of neighborhood characteristics and structure characteristics were roughly equivalent. Although urban environmental characteristics had relatively minimal impacts on housing prices, we cannot neglect the impacts of urban environmental characteristics, which accounted for 16 percent of the total importance. Specifically, the top five characteristics were YEAR (0.4259), EC_DIS (0.3720), C_DIS (0.2494), FH3_NEAR (0.1759) and HPR_DIS (0.1306). For five urban environmental characteristics, the SHAP importance was ranked as follows: UG (0.1145) > UW (0.1043) > SVI (0.0908) > GVI (0.0601) > BVI (0.0570). The SHAP importance of the overhead view environmental characteristics (0.2187)

was slightly higher than those from the horizontal view (0.2079). The horizontal view environmental characteristics could account for 8 percent of total housing prices.

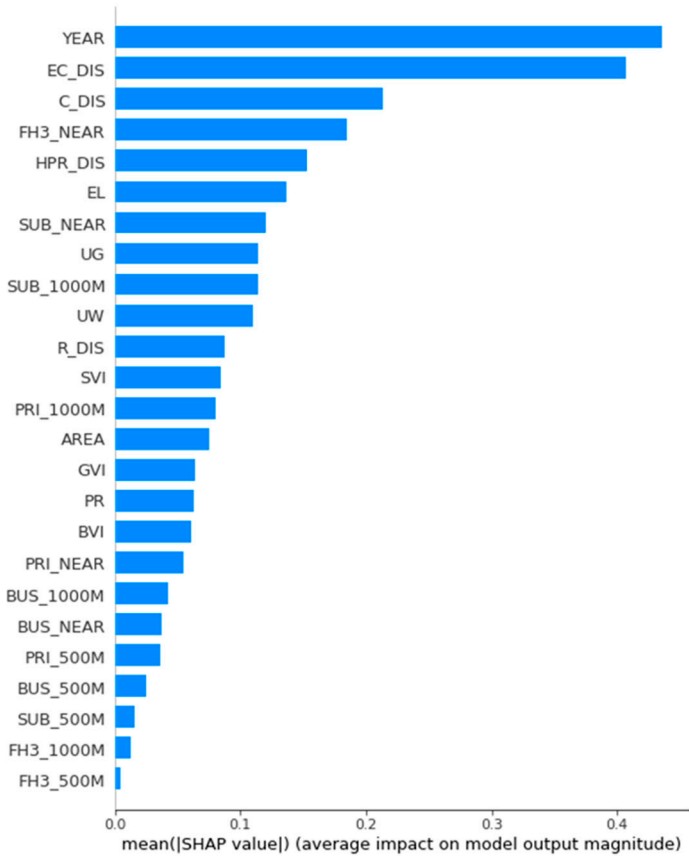

**Figure 8.** SHAP features importance for the determinants of housing prices.

Given that SHAP features importance only contains the absolute value of feature contributions, a density scatter plot of SHAP values for each characteristic was used to further analyze the relationships of the determinants with the housing prices. Characteristics were sorted by the values of SHAP importance. As shown in Figure 9, each point on the summary plot was the Shapley value for a characteristic of a community. The position on the x-axis was determined by the Shapley value, and the color denoted the value from low (blue) to high (red). The dispersion in the y-axis direction represented the number of points, which demonstrated the distribution of the Shapley values per characteristic. If the SHAP value of a characteristic increases with the increase of the corresponding feature value, this characteristic has a positive impact on housing prices, and vice versa. Figure 9 indicates that the four location characteristics all had strong negative relationships with housing prices. YEAR, FH3_NEAR and SUB_NEAR had apparent negative influences on housing prices. EL, SUB_1000M and PRI_1000M showed positive influences on housing prices. In terms of urban environmental characteristics, UW had a strong positive correlation with housing prices. The relationship between SVI and housing prices had a negative correlation. For UG and GVI, although the communities with high SHAP values had relatively high feature values, the SHAP values were not always increased with the increase of the feature values. This result showed that the relationships between these two characteristics and housing prices were complicated and nonlinear. In addition, it was difficult to identify the impacts of BVI on housing prices because of no obvious pattern.

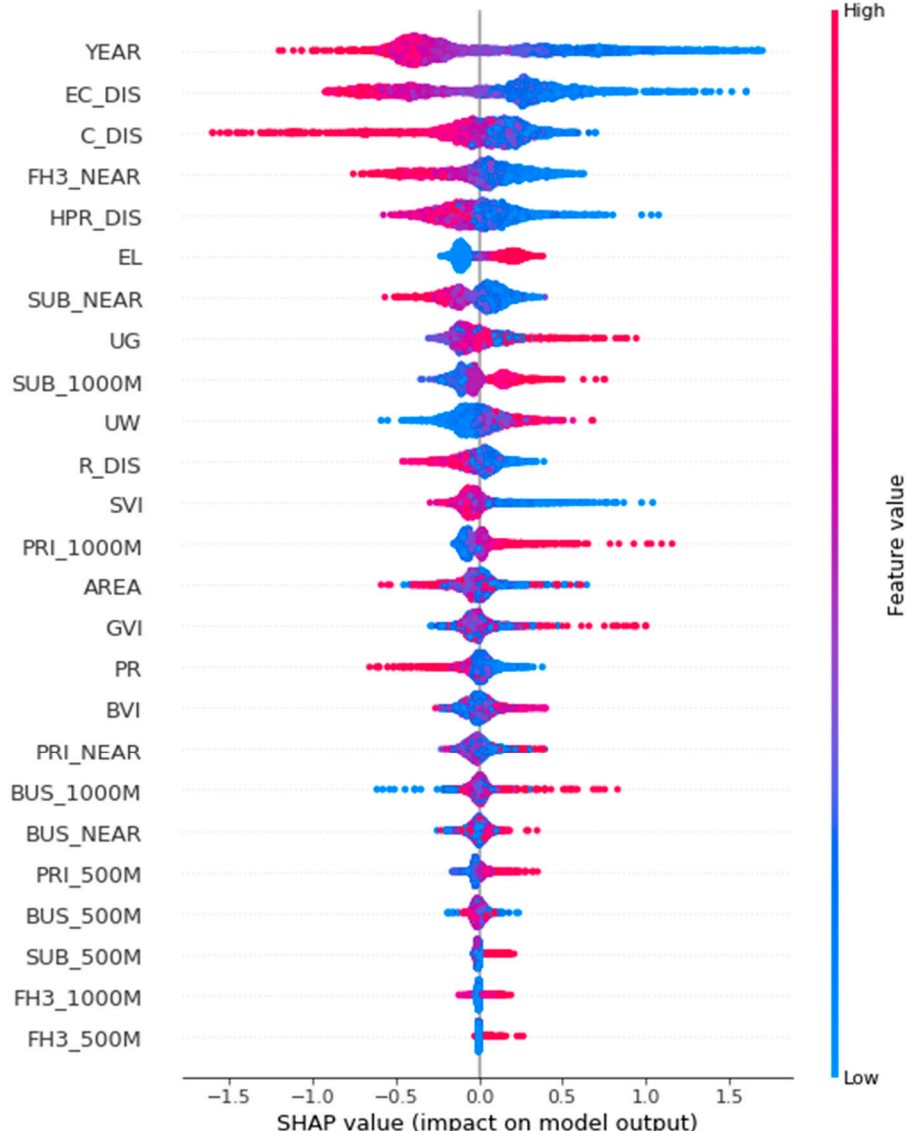

**Figure 9.** SHAP summary plots of housing prices.

### 3.4. Contribution of Urban Environmental Characteristics

Due to that SHAP summary plots couldn't fully reveal the complex and nonlinear relationships between most of the urban environmental characteristics and housing prices, we delved into the specific contributions of characteristics on housing prices by using the SHAP feature dependence plot. The SHAP feature dependence plot for five urban environmental characteristics were drawn in Figure 10 to describe their impacts on housing prices. The spatial distribution of SHAP for the five environmental characteristics were also mapped in Figure 11 to improve the understanding of the contribution of each urban environmental characteristic.

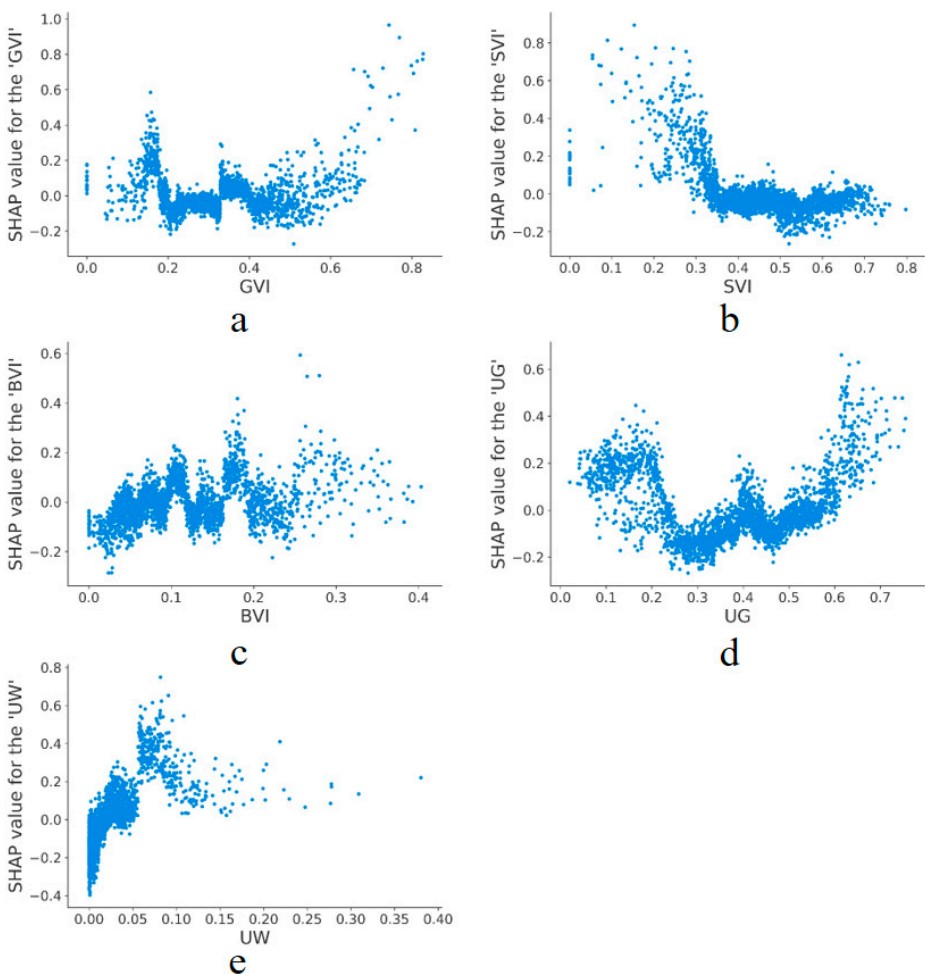

**Figure 10.** SHAP feature dependence plots for the five urban environmental characteristics: (**a**) GVI_SHAP, (**b**) SVI_SHAP, (**c**) BVI_SHAP, (**d**) UG_SHAP and (**e**) UW_SHAP.

### 3.4.1. Contribution of Green View Index (GVI) and Urban Green Coverage (UG)

The SHAP values of the GVI showed a decreasing, stable and increasing tendency, and the two inflection points were approximately 0.2 and 0.5 (Figure 10a). Most of the GVI SHAP values were positive when GVI was less than 0.2 or greater than 0.5. When the GVI exceeded 0.5, the GVI SHAP value increased as the GVI increased. The result of the traditional hedonic model, which was built by the linear regression model, showed that the GVI had a significant positive effect on housing prices (Table 2). Every one percent increase in the GVI can increase housing prices by 71 RMB/m$^2$. Our method indicated that the relationship between the GVI and housing prices was complex and nonlinear rather than linear positive. Shanghai's homebuyers were willing to pay a premium for a green view only when the GVI was of higher value, which was more elaborate than the results of previous studies. A study in the Netherlands showed that a green view can attract an extra price increase of 8% [20]. Another study in Hong Kong also suggested green space views have notably enhanced residential housing prices [23]. To better interpret the results, the spatial distribution of the GVI (Figure 6a) and GVI SHAP (Figure 11a) played an important role. From the distribution of the communities whose GVI and GVI SHAP were both high, we could find that most of these communities were near large parks, such as Changshou Park in the Putuo District, Xujiahui Park in the Xuhui District and Huashan Green Park in the Changning District. These parks could serve as recreational venues and provide pleasant views to residents [54]. The reason why the communities with low GVI values had positive effects on housing prices might be that most of these communities have been built for many years.

Although these older residential communities are lacking a horizontal green view, most of them have diverse public service facilities due to long-term developments.

Compared with the GVI SHAP, a similar trend was observed for the UG SHAP. Figure 10d showed that the SHAP value of the UG was positive when the UG was less than 0.23 and then fluctuated around zero. When the UG was greater than 0.5, the UG SHAP value presented a significant increase. The positive influence of the UG on housing prices when the UG was less than 0.23 or greater than 0.5 indicated that homebuyers were willing to pay more for higher UGs. The reasons for these results were also similar to those reasons for the GVI. Table 2 showed that the GVI was not significant in the traditional hedonic model, which was not consistent with our method. To investigate whether the impacts of the GVI and UG on housing prices show the same pattern, we carried out a comparison between the GVI and UG. The coefficient of determination for the GVI and UG was 0.0799. The spatial distribution of the GVI and UG were quite different. These results suggested that there were no obvious correlations between the GVI and UG. For the SHAP value of the GVI and UG, the coefficient of determination for them was 0.0098. The spatial distribution of the GVI SHAP and UG SHAP were also quite different. Thus, there were no obvious correlations between the GVI SHAP and UG SHAP. All of these results indicated that, although both higher GVI and higher UG had positive impacts on housing prices, there were significant differences between the patterns of their impacts on housing prices. These finding demonstrate that the impacts of the same urban environmental elements from different observation perspectives (horizontal view and overhead view) are different.

In general, the relationships between housing prices and two green characteristics (green view index from street view data and urban green coverage rate from remote sensing) are both nonlinear. Shanghai's homebuyers are willing to pay extra for green only when the green view index or urban green coverage rate are of higher value.

### 3.4.2. Contribution of Sky View Index (SVI)

The SVI of a community could reflect the amount of open spaces, as well as the height and density of buildings in and around this community. In this study, when the SVI value was less than 0.35, the SHAP value of most communities was positive and decreased from 0.8 to zero. For every one percent increase in the SVI, the housing prices decreased by 320 $RMB/m^2$. When the SVI value was greater than 0.35, the SVI SHAP value maintained stable at around zero. The result of the traditional hedonic model showed that the SVI had a significant negative effect on housing prices in Table 2. Every one percent increase in the SVI can decrease housing prices by 123.5 RMB/m2. The findings of our method indicated that the relationship between the SVI and housing prices was also nonlinear rather than linear. By comparing Figures 6b and 11b, we could find the values of the SVI SHAP were the highest in the central area and decreased to the outskirts gradually, which was opposite to the distribution of the SVI. Contrary to expectation, these results mean that the SVI has a strong and negative impact on housing prices in Shanghai when its value is less than 0.35. This finding contrasted with a previous study indicating both street and building views suppressed housing price in Hong Kong [23]. The opposite result in Shanghai could be explained as follows. The high housing prices in Shanghai has resulted in a vertical and compact city, with most residents living in high-density and high-rise residential buildings. The high-rise buildings mean enjoyment of wider views and less noise and air pollution in the higher floors, resulting in a better environmental quality.

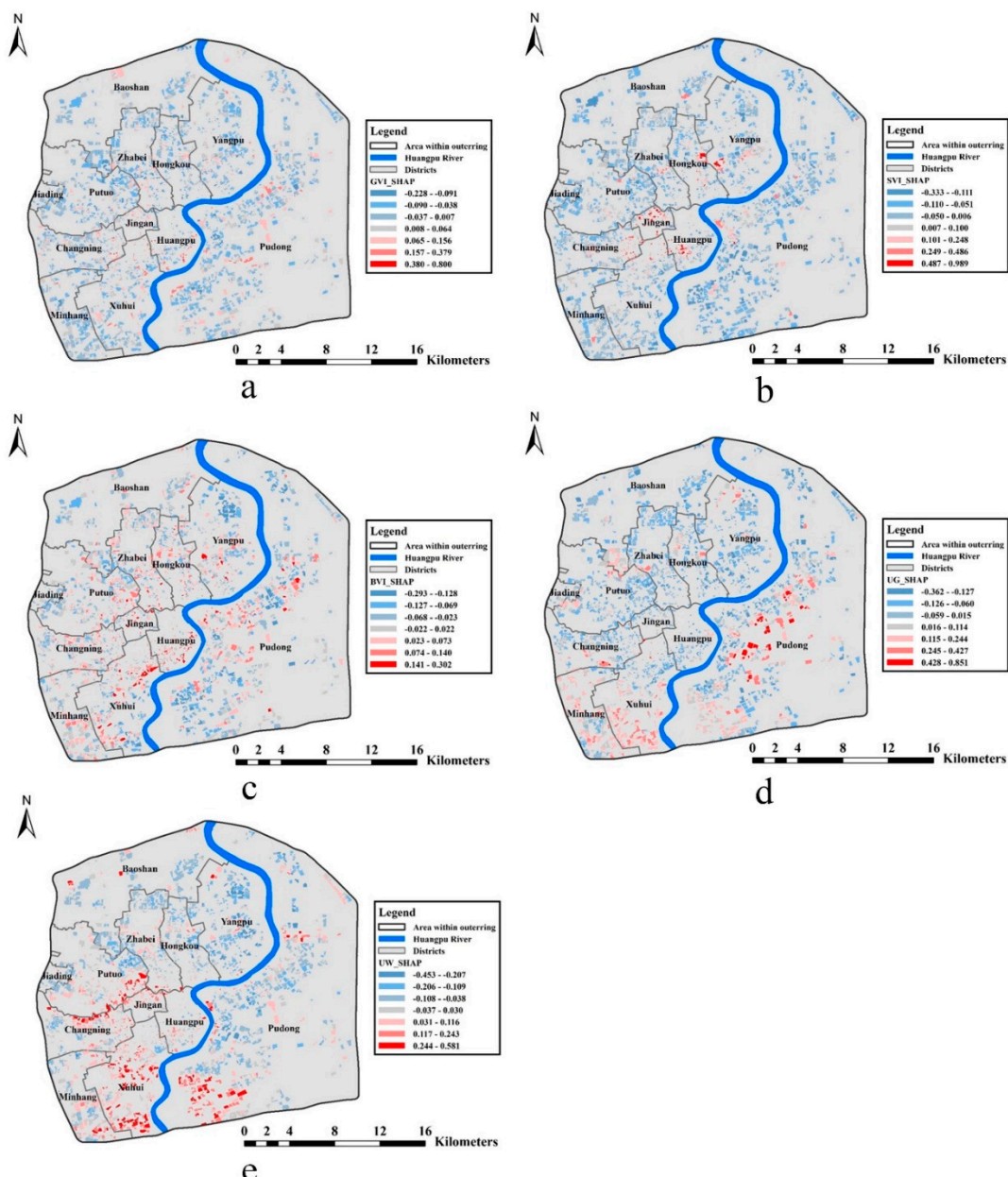

**Figure 11.** Spatial distribution of SHAP for the five urban environmental characteristics: (**a**) GVI_SHAP, (**b**) SVI_SHAP, (**c**) BVI_SHAP, (**d**) UG_SHAP and (**e**) UW_SHAP.

### 3.4.3. Contribution of Building View Index (BVI)

With regards to the BVI, the BVI SHAP value always fluctuated around zero, with a small variance between 0.2 and -0.2. This result demonstrated that the influence of the BVI on housing prices was not obvious. Table 2 showed that the BVI was not significant in the traditional hedonic model, which was consistent with our method. The reason for this result might be that many buildings are blocked by trees and cars in street view images. This leads to how the BVI couldn't depict the distribution of buildings accurately. In most cases, the SVI is the better choice than the BVI for the description of buildings from a horizontal view.

### 3.4.4. Contribution of Urban Water Coverage (UW)

The UW SHAP value increased sharply when the UW was lesser than 0.08, and a one percent increase in the UW SHAP could increase housing prices by 800 RMB/m². When the UW was greater

than 0.08, the UW SHAP value maintained stable. This result indicated that Shanghai's homebuyers would be willing to pay a premium for houses in communities with a higher UW, which was consistent with studies in Hangzhou [55] and Hong Kong [23]. Table 2 showed that the UW was significant and positive in the traditional hedonic model, which is consistent with our method. In spatial distribution, Figures 6e and 11e show that the UW SHAP and the UW presented similar patterns. Communities with a high UW SHAP value were mainly concentrated along the Huangpu River and Suzhou Creek. These two main rivers provide a large amount of water coverage for the communities along them. In a compact city, water bodies have the effect of adjusting air temperature and humidity, which improves human comfort. The water also provides residents with precious spaces where air circulation and solar access are less impeded.

## 4. Conclusions

In this study, we proposed a new framework for measuring the impacts of urban environmental elements on housing prices in the area within Shanghai's outer ring. The green view index (GVI), the sky view index (SVI) and the building view index (BVI) were extracted as horizontal-view urban environmental characteristics based on the Baidu street view images using a deep convolutional neural network. The overhead view environmental characteristics were computed by remote sensing data. Comparing the results of three tree-based ensemble learning models and linear regression models, the XGBoost model showed the best performance. Thereafter, a SHapley Additive exPlanations (SHAP) method, which has the ability to explain the model's overall behavior in the form of particular feature contributions, was introduced to uncover the complex and nonlinear relationships between urban environmental characteristics and housing prices. The spatial distribution of SHAP for the five environmental characteristics were mapped to improve the understanding of the contribution of each urban environmental characteristic. In addition, the impacts of horizontal-view and overhead-view green characteristics on housing prices were compared to analyze the differences of the same urban environmental elements' impacts on housing prices from different observation perspectives. The experimental results are demonstrated as follows. Compared with location, neighborhood and structure characteristics, urban environmental characteristics have relatively minimal impacts that account for 16 percent of housing prices. The relationship between the GVI and housing prices is nonlinear rather than linear positive or linear negative. Similar to the GVI, the urban green coverage rate (UG) also has a nonlinear relation with housing prices. These findings indicated that Shanghai's homebuyers are willing to pay a premium for green only when the GVI or UG are of higher values. Although both a higher GVI and higher UG have positive impacts on housing prices, there are significant differences between their impacts on housing prices. Contrary to previous studies, when the SVI value is less than 0.35, every one percent increase in the SVI, decreases the housing prices by 320 RMB/m2. The potential reason is that high-density and high-rise residential areas often have better living facilities. Compared with the GVI and SVI, the influence of the BVI on housing prices is not obvious. A one percent increase in the urban water coverage rate (UW) can increase housing prices by 800 RMB/m2, which indicates residents in Shanghai are willing to pay a premium for water coverage. In summary, the case of Shanghai shows that the proposed framework is practical and efficient.

This study was limited in several ways. First, the applicability of the proposed framework was tested in Shanghai. Considering the geographical heterogeneity, the relationships between the urban environmental elements and housing prices may be different in a different city. Using this framework to quantify the differences among cities is expected to achieve a promising result. Second, the housing transaction data used in this study were only obtained in 2018. Thus, further studies could be conducted to integrate multi-year data to analyze the temporal dynamics of the impacts of the urban environmental elements on housing prices. Third, our housing model does not consider some housing characteristics, such as floor level and urban village, because these characteristics cannot be captured at present. It is worth discussing these characteristics of the Chinese housing market in future research. Last, the acquisition time of the data for extracting urban environmental characteristics was different.

The Baidu street view data were obtained in 2017, while the remote sensing data were obtained in 2015. Due to the rapid development of Shanghai and seasonal differences of nature environmental elements, differences in data acquisition time could have adverse effects on research findings. Therefore, street view data and remote sensing data with similar acquisition times could be used in future research to improve the results.

**Author Contributions:** Conceptualization, L.C. and X.Y.; methodology, L.C. and X.Z.; software, L.C.; validation, W.C. and T.C.; formal analysis, L.C. and Y.Z.; investigation, L.C.; resources, X.Y.; data curation, L.C. and Y.Z.; writing—original draft preparation, L.C.; writing—review and editing, X.Y., Y.L., W.C., X.Z. and T.C.; visualization, L.C. and supervision, X.Y., Y.L. and T.C. All authors have read and agree to the published version of the manuscript.

**Funding:** This research was funded by the National Natural Science Foundation of China (41701438).

**Conflicts of Interest:** The authors declare no conflicts of interest.

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
