# Peer review of "Measuring Impacts of Urban Environmental Elements on Housing Prices Based on Multisource Data—A Case Study of Shanghai, China"

_ijgi, doi:10.3390/ijgi9020106_

Round 1
Reviewer 1 Report
This study examines the impact of urban environmental elements on housing prices in Shanghai using a Shapley Additive exPlanations method. A number of environmental elements being considered. The results should that the SHAP model is practical and efficiency. This is an interesting study. I like the paper, but it requires some revision.
Major comments:
The model does not consider many key variables such as number of bedroom, number of bathrooms, floor level etc. These are key variables that should be considered in modelling housing prices. The dependent variable is price per square meter. I don’t understand why we include area in the model as the price has been normalised by area. The literature review section can be enhanced. For instance, why we need to control distance to Huangpu River? Further, the authors should discuss the housing submarkets. Recently, Bangura and Lee (2019) in Housing Studies have demonstrated that housing prices behave differently in different housing submarkets. I am wondering how this affect the model. Particularly the Figure 3 shows that Pudong and Huangpu (close to the river) are more expensive than other parts. The methodology: Is there any multicollinearity for distance to the city and distance to the city employment? Urban village- this is a unique housing characteristic in China. How this urban environmental feature would affect the model? See Bi et al. (2019) in Habitat International for the discussion of urban villages and informal housing, as well as residential segregation. If this feature cannot be captured, it should at least be qualified. How the variable of Urban water coverage rate is related with distance to Huangpu River? The authors argued that the sky view index is negative due to high-rise residential area often has better living facilities. Can this be further controlled in the model in order to show it? The non-linear relationship is interesting. A greater discussion is required to explain why residents in Shanghai only willing to pay a premium for green index is high of value. The current discussion sound like a linear link. If non-linear relationship does exist, how about the properties with low green index and medium green index values? It is worth to compare the results from SHAP with the traditional hedonic model? This can see whether the SHAP model provides an enhanced result.Author Response
Please see the attachment

Reviewer 2 Report
In the research, Authors attempted to study the impacts of urban environmental elements on residential housing prices based on multi-source data in Shanghai. The paper has a proper structure and is well-written. However, I cannot recommend it for publishing without major revisions.
The main problem in this study is the data on housing prices. Houses are heterogeneous goods and taking the simple average in the district instead of the price will bias the results (first of all the model should control for characteristics of sold houses – area, technical condition and so on). The hedonic method is based on the assumption that the value of a good depends on its characteristics, so each transaction (price) should be the basis of analysis. Authors should try to control for houses’ characteristics (such as the area, age, exact location, condition). If they do not, the problem of endogeneity will arise (for example bigger houses were sold closer to the rail line in one year, and because of that the average price was higher). The Authors try to examine the influence of GVI, SVI, and BVI on residential prices. However, these indicators may be misleading as the view from different apartments may differ even if the values of GVI, SVI and BVI are the same. This issue should be highlighted. Perhaps it would be better to use vicinity or proximity. The data comes from the web page. The question is if they are asking or transaction prices. Moreover, the time span should be indicated. If more than one year than the time variable should be included in the model. The limitations of this study should be presented. Why is this method better than using simple distances to the urban green areas? Literature review – perhaps some latest results od other studies should be presented – e.g. Journal of Housing and Built Environment, International Journal of Strategic Property Management or Sustainability should be checked.Author Response
Please see the attachment.

Round 2
Reviewer 1 Report
This revised paper has addressed my comments satisfactorily. However, I still have some hesitation on several parts.
I acknowledged the authors did not have the data of floor level etc. Please acknowledge this as a limitation of data for this study, and this might alter the results. Point 2- please explain this in the paper. Why the normalisation of price per square meter does not fully capture the housing price dynamics? As such, the variable of area should be introduced. Point 3- OK Point 4- OK. I would suggest that the author further discuss it. See my suggestion:“The explanatory variables of housing prices have been widely discussed in the housing literature. Bangura and Lee (2019), Lee et al. (2017) in IJHMA and Al-Masum and Lee (2019) in IJHMA discussed the ability of market fundamentals in explaining housing prices from the macroeconomic perspective, whilst Trojanek, R. and Gluszak (2018) examined the importance of housing attributes in explaining housing prices from the microeconomic perspective.”
OK I suggested the authors to qualify it and acknowledge it as a limitation of data as this cannot be captured, but it is worth for future study as Bi et al. (2019) have discussed this is a unique characteristic of Chinese housing market. OK OK OKAuthor Response
Please see the attachment.

Reviewer 2 Report
The Authors have improved the article significantly. I recommend the article for publication. The paper is well-written, is a detailed account of the literature review and has the proper structure. This approach is in the meanwhile state of the art. I find that that the paper is methodologically sound, and finds significant results that have direct relevance to academic audiences. I think the manuscript is suitable for publication and this study will contribute to the literature.
